# MiR-27b-3p Reduces the Efficacy of Propranolol in the Treatment of Infantile Hemangioma by Inhibiting the Expression of Apaf-1

**DOI:** 10.3390/biomedicines13051092

**Published:** 2025-04-30

**Authors:** Jing Li, Yifei Zhu, Linyang Xie, Sina Ahmadi, Chonghao Yao, Hao Cui, Xuteng Kang, Junbo Tu, Sijia Na

**Affiliations:** 1Key Laboratory of Shaanxi Province for Craniofacial Precision Medicine Research, College of Stomatology, Xi’an Jiaotong University, Xi’an 710004, China; 2Department of Oral and Maxillofacial Surgery, College of Stomatology, Xi’an Jiaotong University, Xi’an 710004, China

**Keywords:** infantile hemangioma, propranolol, miR-27b-3p, Apaf-1, apoptosis

## Abstract

**Objective**: To explore the role and mechanism of miR-27b-3p in treating infantile hemangiomas (IHs) with propranolol and to clarify the cause of the poor efficacy of propranolol in IHs. **Methods**: Human umbilical vein endothelial cells (HUVECs) were used as the research model and were treated with 0, 15, 30, 45, 60, and 90 μM of propranolol to explore the best concentration. RNA interference technology was used to regulate the expression of miR-27b-3p. CCK-8, TUNEL, and flow cytometry detected cell proliferation and apoptosis levels. Real-time PCR was used to detect the expression of miR-27b-3p and apoptosis pathway-related mRNA, and Western blotting was used to detect the expression of apoptosis-related proteins. The target relationship between miR-27b-3p and Apaf-1 was analyzed using a double Luciferase report. **Results**: The most significant inhibitory effect on cell activity of propranolol is at a dose of 30 μM. After propranolol treatment, the expression of miR-27b-3p was downregulated, and the expression of the apoptotic factors Apaf-1, PARP, caspase-9, and caspase-3 was upregulated, which was consistent with the results after the deletion of miR-27b-3p. However, after upregulation of miR-27b-3p, the level of and the expression of apoptotic factors was inhibited. “targetscan.org” gene database analysis found that miR-27b-3p matched the 3′-UTR of Apaf-1 mRNA, and luciferase results showed that miR-27b-3p had a targeted relationship with Apaf-1. **Conclusions**: The miR-27b-3p target inhibits the expression of Apaf-1, reduces the level of endothelial cell apoptosis, and interferes with the therapeutic effect of propranolol.

## 1. Introduction

Infantile hemangiomas (IHs) are the most common benign tumors of vascular origin in infants and children, with an incidence of about 3–10% and commonly occurring at birth or within the first month of life [1]. However, some IHs still proliferate, destroy tissues, and impair function, resulting in long-term residual permanent skin damage that can seriously affect children’s physical and mental health and even threaten their lives [2,3].

Oral propranolol was first reported for treating IHs in 2008 and has subsequently been widely used and become the first-line drug for treating IHs [4]. Propranolol may promote IH regression by vasoconstriction, inhibiting angiogenesis, and inducing apoptosis [5]. However, there is no universal recognition of the mechanism of propranolol for its treatment [6]. Our previous studies have shown that propranolol accelerates IH regression by promoting hemangioma endothelial cell apoptosis [7,8,9]. Moreover, apoptosis is a series of continuous and multi-gene-regulated processes. Apoptotic protease activating factor-1 (Apaf-1) is key in the mammalian mitochondrial-dependent apoptosis pathway. Apaf-1 binds to cytochrome c and dATP and cleaves the precursor protein, releasing and activating caspase 9, further inducing the caspase cascade and promoting apoptosis [10,11].

MicroRNAs (miRNAs) are small, noncoding RNAs that play important roles in many biological processes, such as cell proliferation, apoptosis, tumor angiogenesis, and so on [12]. Previous studies have confirmed that many miRNAs are involved in the development of infantile hemangiomas [13]. For example, miR-126 is a positive regulator of angiogenic signaling and vascular endothelial integrity [14]. miR-424 interacts with VEGFA (vascular endothelial growth factor A) in senile hemangiomas [15]. Moreover, miR-125b interacts with VEGFA to regulate angiogenesis [16]. However, the potential mechanism of miR-27b-3p’s involvement in the development of infantile hemangiomas remains unclear. It has been reported that miR-27b-3p is associated with reducing the malignancy of hemangioma-derived endothelial cells. IL-10 upregulates miR-27b-3p, which inhibits PCNA (a marker of cell proliferation) by destabilizing its mRNA and inhibiting the transcription factor YY1. This pathway limits endothelial cell proliferation in IHs [17]. It was reported that the β-adrenoceptor regulates miRNA expression in rat hearts [18]. Thus, whether propranolol exerts its proliferation, inhibition, and apoptosis-promoting effects on infantile hemangioma cells by regulating the expression of miR-27b-3p needs to be further investigated. miRNAs typically recognize binding sites in the 3′-untranslated region (3′-UTR) of mRNA transcripts [19]. They regulate the activity of target genes through complementary pairing with the 3′-UTR region of target mRNA after transcription, thereby regulating different biological processes [20]. Previous studies have found that miR-23a/b and miR-27a/b can effectively inhibit the expression of Apaf-1 in the central nervous system [21]. It was found that miR-27b-3p is complemented with the mRNA3′-UTR region of Apaf-1 through “targetscan.org” gene database analysis. Recent studies have reported that circRNF 111 regulates gastric cancer cells’ growth, migration, and invasion by binding to miR-27b-3p [22]. However, whether circRNF 111 increases the sensitivity of propranolol to IH treatment through the miR-27b-3p/Apaf-1 axis remains to be determined.

In this study, we first explored propranolol’s effects on HUVEC activity and apoptosis. Then, we demonstrated that the knockdown of miR-27b-3p promotes propranolol-induced apoptosis in HUVECs. Finally, we studied the effect of the circRNF 111/miR-27b-3p/Apaf-1 axis on propranolol-induced apoptosis in HUVECs.

## 2. Materials and Methods

### 2.1. Cell Culture and Treatment

HUVECs were purchased from Sebacon and frozen in liquid nitrogen at −196 °C. They were cultured in α-MEM (BOSTER, Wuhan, China) with 20% fetal bovine serum (Gibco, Vacaville, CA, USA) at 37 °C in a humid atmosphere with 5% CO_2_.

Propranolol hydrochloride was dissolved in PBS at pH 7.4 away from light. The concentration of the mother liquor was adjusted to 1000 μM, and the pH to 6.0 for reserves. HUVECs were treated with different concentrations of propranolol (0, 15, 30, 45, 60, and 90 μM) at 70% confluence.

### 2.2. Cell Transfection

HUVECs were planted in 6-well plates until they reached 70% cell confluence using HitransG A, and the transfection reagent was transfected according to product specifications. The miR-27b-3p knockdown group (miR IN group) and the negative control group (miR NC group) were obtained by transfecting LV-has-miR-27b-3p-inhibition and its negative control (CON137), which were synthesized by Genechem Co., Ltd., Shanghai, China. LV-has-miR-27b-3p mimics and negative controls were synthesized by Shanghai Genechem Co., Ltd., who transfected the established miR-IN group, the miR-IN + miR-mimics NC group, and the miR-IN + miR-mimics group using a GP-transfection reagent transfected according to product specifications. To overexpress circRNF 111, we transfected cells with LV-circRNF 111 and a negative control designed and synthesized by GenePharma (Shanghai, China). This led to the creation of the circRNF 111 overexpression group (LV-circRNF 111) and the negative control group (LV-NC), with transfection carried out using HitransG P (Genechem Co., Ltd., Shanghai, China).

### 2.3. Cell Counting Kit-8

HUVECs were inoculated on 96-well plates at a rate of 5000/well. The cells were treated with propranolol hydrochloride with concentration gradients of 0, 15, 30, 45, 60, and 90 μM when the cells grew to 70%. Cell activity was detected using Cell Counting Kit-8 (CCK-8) (BOSTER, Wuhan, China). After propranolol treatment, the CCK-8 solution was briefly added to the complete medium. Cells were incubated for 1 h at 37 °C in a humid atmosphere with 5% CO_2_. The absorbance was measured at 450 nm using a Microplate Reader (Bio-Rad, Hercules, CA, USA).

### 2.4. Flow Cytometry

Cells were seeded in 6-well plates at a density of 1 × 10^5^ cells per well. Apoptosis was assessed using the Annexin V-APC/7-AAD apoptosis detection kit (MU LTISCIENCES, Hangzhou, China). The protocol involved collecting 2 × 10^5^ cells after discarding the supernatant, followed by three washes with chilled phosphate-buffered saline. The cells were then centrifuged, resuspended in 500 μL of diluted Annexin V Binding Buffer, and stained with 5 μL each of Annexin V-APC and 7-AAD. After gentle mixing, the cells were incubated in the dark at room temperature for 15 min. Flow cytometry was conducted within an hour using a Becton Dickinson FACScan, with data analyzed using FlowJo 10.4 software.

### 2.5. TUNEL Staining

The apoptotic level was evaluated using a TUNEL kit (BOSTER, Wuhan, China). HUVECs were inoculated in confocal dishes with 200 μL medium per well at a density of 2000 cells per well and cultured in a 5% CO_2_ incubator at 37 °C for 24 h. Then, propranolol hydrochloride (experimental group) or PBS (control group) was added for 24 h. Washing was performed with pre-cooled 0.01 M PBS three times, and the supernatant was discarded. HUVECs were first fixed in 4% formaldehyde for 10 min, and mixed with 0.5% Triton X-100 for 10 min after being washed several times with 0.01 M PBS to break the membrane. After several washes with 0.01 M PBS, proteinase K was diluted with 0.01 M TBS at a 1:200 ratio and added to the dishes for digestion at 37 °C for 10 min, followed by three washes with TBS. A TUNEL reaction mixture, consisting of 18 μL of buffer, 1 μL of TdT, and 1 μL of DIG-d-UTP, was added to each dish at a rate of 20 μL per dish and incubated at 37 °C for 2 h, followed by three washes with TBS. A blocking solution was then added at 50 μL per dish and incubated at room temperature for 30 min before drying. A biotinylated anti-digoxin antibody was diluted with a SABC diluent at a 1:100 ratio and 50 μL per tablet, incubated at 37 °C for 30 min, and washed with TBS 3 times. SABC was diluted with a SABC diluent at a 1:1000 ratio and 50 μL per tablet, incubated at 37 °C for 30 min, and washed with TBS 3 times. Finally, the DAPI dye solution was re-dyed, double-steamed water was cleaned twice, the anti-fluorescence quench agent was sealed, and a laser confocal microscope was used for observation, recording, and analysis.

### 2.6. RNA Extraction and Quantitative RT-PCR

RNA was isolated, collected, purified, and extracted from HUVECs according to the operating instructions of the Trizol Kit (Accurate Biology, Changsha, Hunan, China). According to the instructions, cDNA was obtained by reverse transcription, and miRNA was obtained using the miRNA first strand cDNA synthesis kit (AG11716, Accurate Biology, Changsha, Hunan, China). circRNA and mRNA were treated with the Evo M-MLV RT Mix Kit with gDNA Clean for qPCR (AG11728). The SYBR^®^ Green Premix Pro Taq HS qPCR Kit (AG11701) and SYBR^®^ Green Premix Pro Taq HS qPCR Kit II (AG11702) were then used for RT-qPCR experiments. PCR reaction conditions: 95 °C for 30 s, then 39 cycles of 95 °C for 5 s, 60 °C for 20 s, and finally 95 °C for 10 s. The specific primer sequences are shown in Table 1.

### 2.7. Western Blotting

Protein extraction was carried out according to the RIPA cracking solution kit (Zhonghui Hecai, Xi’an, China), protein quantification was performed with the BCA protein analysis kit (BOSTER, Wuhan, China), and the gel was prepared with the PAGE gel (color gel) rapid matching kit (Zhonghui Hecai, Xi’an, China). After water bath denaturation, the protein was isolated by SDS electrophoresis, transferred to a PVDF membrane, and sealed with 5% skim milk powder TBST for two hours. Then, anti-Apaf-1 (1:1000; Abcam; cat.no.ab234436, Shanghai, China), anti-PARP (1:1000, BOSTER, Wuhan, China, bs-2138R), anti-Caspase-9 (1:5000; Abcam; cat.no.ab32539), anti-Caspase-3 (1:5000; Abcam; cat.no.ab32351), anti-cleaved-Caspase-3 (1:500; Abcam;cat.no.ab2302), and anti-GAPDH (1:5000, BOSTER, Wuhan, China) were added. Samples were incubated overnight at 4 °C. The next day, the sheep anti-rabbit IgG-HRP (1:5000; BOSTER, Wuhan, China) was incubated at room temperature for two hours, and then 200 μL of luminous solution was added. The samples were then loaded onto the machine (Bio-Rad ChemiDoc XRS system, Life Science, Shanghai, China). Quantitative analysis was conducted using Image Lab software (12012931, Life Science, Shanghai, China).

### 2.8. Luciferase Reporter Assay

circRNF 111, Apaf1-3′UTR, and their miR-27b-3p binding site mutant sequences were synthesized and added to the luciferase reporter vector psiCHECK2 (Hanbio, Biotechnology, Wuhan, China). They are named h-RNF111-3UTR-wt and h-RNF111-3UTR-mu and h-APAF1-3UTR-wt and h-APAF1-3UTR-mu, respectively. 293T cells were then co-transfected with hsa-miR-27b-3p, a negative control (NC). Following the manufacturer’s protocols of the Dual Luciferase Assay Kit (Hanbio, Biotechnology, Wuhan, China), the relative luciferase activity was examined using a full-wavelength multifunctional enzyme label tester (SpectraMax M5e) (Molecular Devices, Shanghai, China).

### 2.9. Statistical Analysis

All experiments were repeated 3 times, and the results were expressed as means ± standard deviation (SD). SPSS Vision 27.0 (SPSS, Chicago, IL, USA) was used for statistical analysis. A Student’s *t*-test or one-way ANOVA was used to calculate the *p*-values. A *p*-value less than 0.05 was considered a statistically significant difference.

## 3. Results

### 3.1. Propranolol Effectively Inhibited the Activity of HUVECs and Promoted Apoptosis In Vitro

Firstly, we examined HUVEC viability at 0, 15, 30, 45, 60, and 90 μM of propranolol. CCK-8 results showed that the activity of the propranolol-treated group was significantly inhibited (Figure 1A). The most significant decrease in HUVEC viability was observed in the 30 μM group, which was used for the follow-up experiment. Subsequently, HUVECs were treated with propranolol at a concentration of 30 μM for 1, 2, 3, 4, and 5 days. Moreover, the results of CCK-8 experiments showed that cell activity was gradually inhibited with increasing time (*p* < 0.05 or *p* < 0.001, Figure 1B). Compared with the control group, flow cytometry and TUNEL staining showed that the apoptosis of HUVECs in the propranolol-treated group, respectively, increased by 71.02% (*p* < 0.001, Figure 1C) and 11 times (*p* < 0.001, Figure 1D). Figure 1E,F show that the mRNA and protein expression levels of Apaf-1, PARP, caspase-9, caspase-3, and cleaved cas3 were significantly upregulated after HUVECs were treated with propranolol for 24 h (*p* < 0.001). Therefore, we found that propranolol promoted HUVEC apoptosis and inhibited proliferation in vitro.

### 3.2. Propranolol Promotes HUVEC Apoptosis via Downregulating miR-27b-3p

Firstly, the expression level of miR-27b-3p in HUVECs was found to be reduced, as measured by assays, after propranolol treatment (*p* < 0.01, Figure 2A). Subsequently, miR-27b-3p expression was knocked down after transfecting HUVECs with the miR-27b-3p inhibitor. Figure 2B shows that the stable strains of HUVECs with miR-27b-3p knockdown were successfully established, identified, and compared with the miR-NC group. The CCK-8 assay, flow cytometry, and the TUNEL assay confirmed that the activity of HUVECs in the knockdown of miR-27b-3p was effectively inhibited after propranolol induction. Apoptosis levels increased by 10.8% (flow cytometry) and 19.19-fold (TUNEL staining) (*p* < 0.001, Figure 2C–E). Similarly, the mRNA and protein levels of Apaf-1, PARP, caspase-9, caspase-3, and cleaved cas3 in the propranolol + miR-IN group were significantly upregulated (*p* < 0.01 or *p* < 0.001, Figure 2F,G). Therefore, we inferred that propranolol promotes HUVEC apoptosis by downregulating miR-27b-3p expression.

### 3.3. Targeted Inhibition of Apaf-1 by miR-27b-3p Reduces Propranolol-Induced Apoptosis in HUVECs

To investigate the molecular mechanism of the propranolol-induced downregulation of miR-27b-3p to promote HUVEC apoptosis, the 3′UTR match of miR-27b-3p with Apaf-1 mRNA was analyzed and predicted by the “targetscan.org” database (Figure 3A). In addition, according to the dual luciferase reporter assay, the activity of the luciferase reporter vector carrying the Apaf-1 3′UTR-WT sequence could be significantly decreased by miR-27b-3p mimics (*p* < 0.001, Figure 3B). We then performed salvage experiments under propranolol treatment by transfecting miR-27b-3p mimics and further clarified that miR-27b-3p inhibited propranolol-induced apoptosis in HUVECs by decreasing Apaf-1 expression. The RT-qPCR results showed that compared with the IN group, the mRNA expression of apoptosis-related factors in the miR-27b-3p mimics transfection group was significantly downregulated (*p* < 0.001, Figure 3C). Meanwhile, Western blotting results were consistent with PCR results, showing that the expression levels of apoptosis-related factors were also significantly downregulated (*p* < 0.01 or *p* < 0.001, Figure 3D,E). These results suggest that miR-27b-3p reduces the propranolol-induced apoptosis of HUVEC cells by targeting Apaf-1 inhibition.

### 3.4. Propranolol Upregulates circRNF 111 to Promote HUVEC Apoptosis via miR-27b-3p/Apaf-1

circRNA is mainly used as a miRNA “isolator” to inhibit miRNA activity, and the ENCORI database predicts that circRNF111 has a complementary binding domain to miR-27b in the chr15: 59, 323, 361-59, 323, and 582 regions (Figure 4A). Subsequently, miR-27b-3p luciferase reporter vector experiments were designed with wild-type (WT) or mutant (Mut) circRNF 111 binding sites. Compared with the NC group, hsa-miR-27b-3p significantly downregulated the luciferase expression of h-RNF111-3UTR-WT (*p* < 0.001), but hsa-miR-27b-3p did not downregulate the luciferase expression of h-RNF111-3UTR-MUT (*p* > 0.05). There may be a direct interaction between circRNF111 and miR-27b-3p (Figure 4B). circRNF 111 expression was upregulated after HUVEC induction by propranolol (*p* < 0.001, Figure 4C). Subsequently, stable strains of HUVECs in the circRNF 111 overexpression group (LV-circRNF 111 group) and the negative control group (LV-NC group) were established and identified successfully (*p* < 0.001, Figure 4D). Figure 4E showed that miR-27b-3p expression in HUVECs was downregulated after circRNF 111 overexpression. In addition, after treatment with propranolol, compared with the LV-NC group, the expression of miR-27b-3p in the LV-circRNF 111 group was further downregulated (*p* < 0.05, *p* < 0.001). The mRNA and protein expression levels of Apaf-1, PARP, caspase-9, and caspase-3 were significantly upregulated (*p* < 0.001, Figure 4F,G). Therefore, circRNF 111 can be used as an isolator of miR-27b-3p to regulate the translation expression of Apaf-1, thereby promoting the apoptosis of HUVECs induced by propranolol.

## 4. Discussion

An IH is a benign skin tumor caused by excessive proliferation of vascular endothelial cells in the embryonic period. The treatment of IHs usually begins in the early stages of tumor proliferation, when many treatments are available, of which propranolol is recognized as the first-line drug for treating IH. As a non-selective β-adrenergic receptor antagonist, propranolol can block the effects of adrenaline and norepinephrine. Clinically, systemic administration is used to treat children with infantile hemangiomas. Propranolol inhibits the activation of βARs and regulates their downstream signal transduction pathways, such as cAMP, VEGFR2, PI3K-AKT-mTOR, and MAPK, thereby inhibiting the proliferation of hemangioma endothelial cells and inducing apoptosis [23]. However, existing studies are not sufficient to explain the molecular network mediated by propranolol, and the molecular mechanism of propranolol in the treatment of IHs is not fully understood. Therefore, we used the HUVEC model to explore the molecular mechanism of propranolol in the treatment of IHs in vitro.

Previous studies have found that propranolol promotes the apoptosis of hemangioma endothelial cells through the p53-BAX mitochondrial apoptosis pathway, thereby accelerating IH regression [7,9]. Apaf-1 is a key factor in endogenous and mitochondrial-dependent apoptotic pathways and plays a key role in programmed cell death in multicellular organisms. Many studies have reported that in various malignant tumors, activated Apaf-1 can increase the sensitivity of chemotherapy drugs and promote apoptosis of tumor cells. In the present study, by setting the time gradient, we found that propranolol effectively inhibited HUVEC activity in a concentration-dependent and time-dependent manner as time increased. Moreover, at the same time, flow cytometry and TUNEL experiments showed that propranolol promoted HUVEC apoptosis. In this study, the expression of apoptosis-related factors such as Apaf-1 in propranolol-treated HUVECs was downregulated, indicating that propranolol activates the Apaf-1-mediated apoptosis pathway and induces HUVEC apoptosis. There is a competitive endogenous RNA (ceRNAs) regulatory network between circRNA, miRNA, and mRNA, which is related to regulating tumor development.

For example, circRNA_0025202 regulates tamoxifen sensitivity and tumorigenesis in breast cancer by modulating the miR-182-5p/FOXO3a axis; as a miR-326 sponge, circ_0000515 promotes cervical cancer progression by upregulating ELK1 [24,25]. Circ-RNF111 aggravates gastric cancer malignancy through mir-876-3p-dependent KLF12 regulation [26]. CircAP2A2 adsorbs miR-382-5p and regulates VEGFA expression to participate in the progression of infantile hemangioma [27]. However, there are few studies on ceRNA in IHs, and the mechanism of ceRNA in regulating IHs is still in the exploratory stage. As a noncoding RNA, miRNA plays an important role in laryngeal cancer and inhibits norepinephrine-induced cardiomyocyte apoptosis by regulating the expression of Apaf-1 [28]. In recent years, studies have reported that miRNA can regulate angiogenesis, including vascular sprouting, vascular endothelial cell proliferation, survival, migration, and recruitment of vascular progenitor cells [29]. By controlling the expression of angiopoietin and promoting hemangioma endothelial cell apoptosis, which is involved in the maturation of new blood vessels, miR-27b-3p is a subtype of the miR-27 family, which is involved in the development of gastric cancer, gliomas, breast cancer, and other diseases [30,31,32]. MiR-27b-3p can promote the apoptosis of hypoxic neurons by inhibiting Apaf-1, and this inhibition gradually decreases with age, similar to some infantile hemangiomas that gradually fade with age. CircRNA is also a functional regulator in regulating cell differentiation, angiogenesis, immune responses, inflammatory responses, and carcinogenesis [33]. Interestingly, a recent study found that circRNA is upregulated under hypoxic or normoxic conditions and controls endothelial cell nodule formation and spherical budding [34]. CircRNF111 is considered to be closely related to the occurrence and development of breast cancer, colorectal cancer, and gastric cancer. Furthermore, it has been confirmed that hypoxia is involved in the development of IHs, and that hypoxia is driven by HIF-2α. It is well-known that HIF-2α levels are regulated by inflammation, differentiation, and stress signaling [35]. If we can develop specific targeting agents for any step of the pathway, then propranolol will have a better effect on the treatment of hemangiomas.

However, reports on CircRNF111 in IHs are lacking, and its function and mechanism are still unclear. This study’s bioinformatics prediction and dual luciferase assay confirmed that miR-27b-3p had a good binding relationship with Apaf-1 mRNA, circRNF111, and miR-27b-3p. Subsequently, miR-27b-3p knockdown, stable transfection, and propranolol induction were constructed, and the mRNA and protein levels of apoptosis-related factors were upregulated. The rescue experiment reversed the above results, indicating that miR-27b-3p targeted the inhibition of Apaf-1 and reduced propranolol-induced HUVEC apoptosis. In order to further verify the role of circRNF111 in propranolol-induced HUVEC apoptosis, after propranolol treatment, overexpression of circRNF111 in HUVECs further downregulated the expression of miR-27b-3p and promoted the expression of apoptosis-related factors. In summary, circRNF111 binds to miR-27b-3p as a ceRNA and competitively inhibits miRNA binding to Apaf-1 mRNA, thereby increasing Apaf-1 protein expression and promoting the apoptosis of hemangioma endothelial cells. It is suggested that propranolol promotes HUVEC apoptosis through the circRNF 111/miR-27b-3p/Apaf-1 ceRNA network. However, the limitation of our research is the lack of in vivo experiments. The conclusion would be more accurate if there were in vivo experiments. Last but not least, propranolol treatment is now the first drug of choice for IHs. However, lesion relapse after therapy remains an open question. To convincingly answer this question, more research is warranted at the population level that can be further integrated and guide personalized treatment for children who experience side effects or who are more susceptible to propranolol.

This study reveals that propranolol treats hemangiomas by activating the APAF1-mediated apoptotic pathway, while circRNF 111 induces propranolol-induced HUVEC apoptosis via miR-27b-3p/Apaf-1. This will deepen the understanding of the molecular regulatory mechanism of noncoding RNA in the pathophysiological process of hemangiomas and provide a theoretical basis for the sensitization effect of propranolol in treating hemangiomas. If there is a targeted inhibitor of miR-27b-3p, propranolol can be prompted to induce apoptosis by reducing the expression of miR-27b-3p, which has an important role in the treatment of hemangiomas. This is also the first report on the role of miR-27b-3p in the treatment of hemangiomas with propranolol. These results will deepen our understanding of propranolol in the treatment of infantile hemangiomas and provide a theoretical basis for the sensitization effect of propranolol in treating hemangiomas.

## Figures and Tables

**Figure 1 biomedicines-13-01092-f001:**
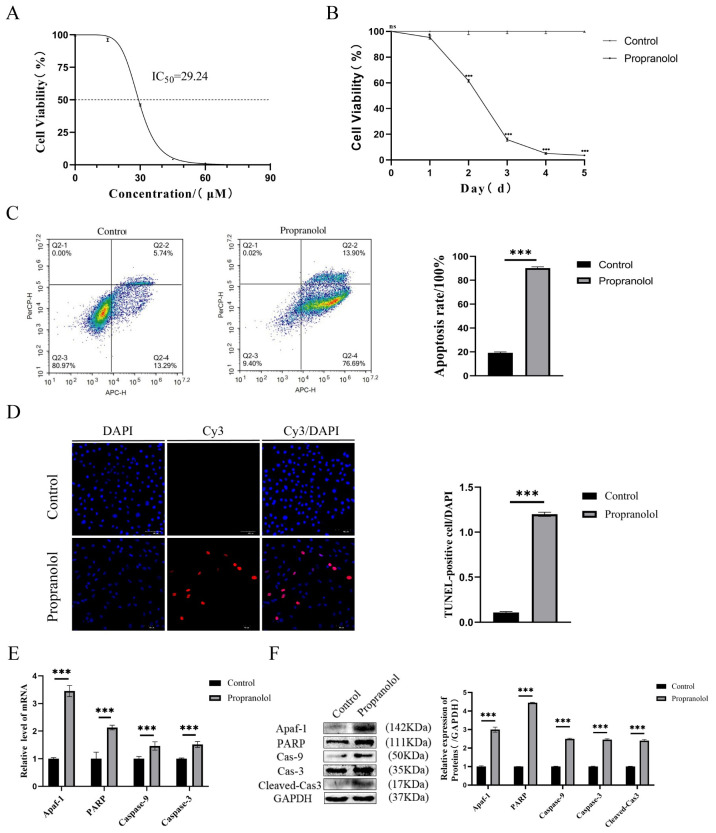
The effect of propranolol on HUVEC activity and apoptosis in vitro. (**A**) CCK8 assay showed that propranolol induced HUVEC apoptosis in a concentration-dependent manner (IC50 = 30 μM). (**B**) The CCK8 assay showed that propranolol hydrochloride (concentration = 30 μM) induced endothelial cell apoptosis in a time-dependent manner. (**C**) Flow cytometry showed that apoptosis after propranolol treatment was higher than in the control group. (**D**) The results of TUNEL staining showed that apoptosis in HUVECs increased after treatment with propranolol hydrochloride, 20× total magnification. (**E**,**F**) After the introduction of propranolol, the mRNA and protein levels of Apaf-1, PARP, caspase-9, caspase-3, and cleaved cas3 in HUVECs were increased. * *p* < 0.05; ** *p* < 0.01; *** *p* < 0.001; ns means no statistical significance.

**Figure 2 biomedicines-13-01092-f002:**
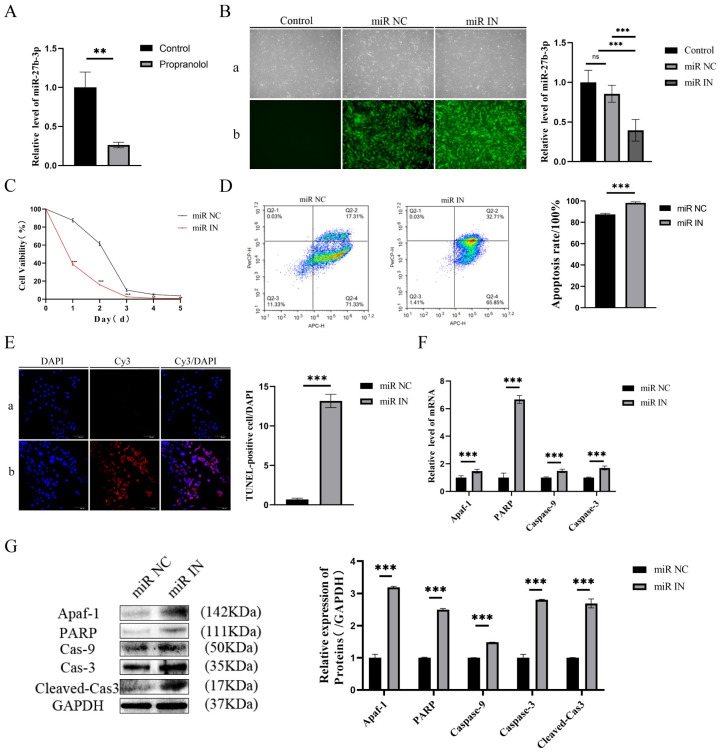
The influence of miR-27b-3p on the apoptosis of HUVECs treated with propranolol. (**A**) qRT-PCR showed that the expression of miR-27b-3p in HUVECs was downregulated after 24 h of treatment with propranolol. (**B**) QRT-PCR verified the inhibition efficiency of miR-27b-3p inhibition (a: light mirror; b: fluorescence microscope), 4× total magnification. (**C**) The CCK8 results demonstrate that under the induction of propranolol, the HUVEC activity of the miR-27b-3p knockdown was inhibited. (**D**) Flow cytometry showed that the knockdown of miR-27b-3p promoted apoptosis of HUVECs due to propranolol. (**E**) TUNEL staining (a: miR NC group; b: miR IN group) showed that knockdown of miR-27b-3p promoted apoptosis in HUVECs treated with propranolol, 20× total magnification. (**F**,**G**) After the knockdown of miR-27b-3p, the mRNA and protein expression levels of Apaf-1, PARP, caspase-9, caspase-3, and cleaved Cas3 were further upregulated. * *p* < 0.05; ** *p* < 0.01; *** *p* < 0.001; ns means no statistical significance.

**Figure 3 biomedicines-13-01092-f003:**
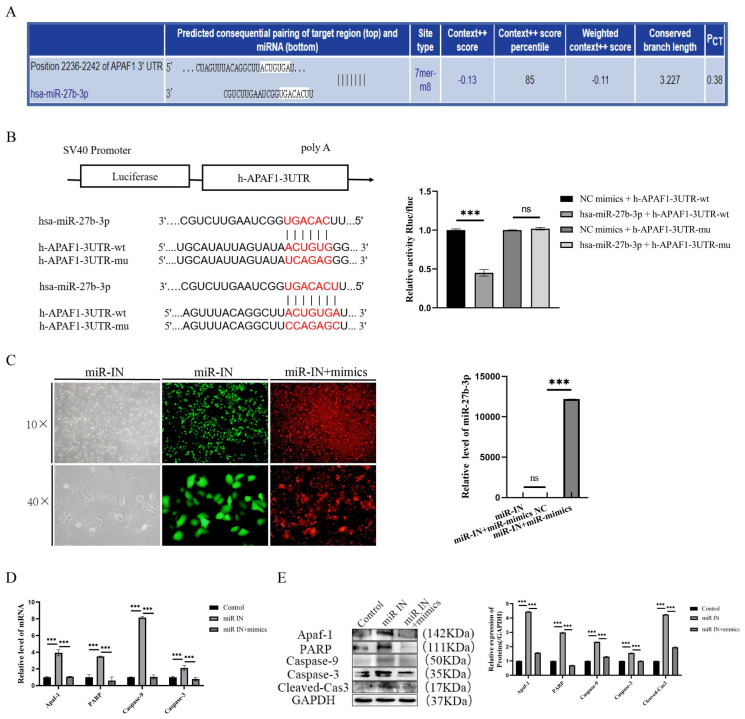
Apaf-1 is a target of miR-27b-3p. (**A**) The targetscan.org database shows that miR-27b-3p complements the Apaf-1 mRNA3′-UTR region. (**B**) Targeted relationship between miR-27b-3p and Apaf-1; luciferase reporter gene assay confirmed the targeted relationship between miRNA-27b-3p and Apaf-1. (**C**) qRT-PCR showed the successful transfection of miR-27b-3p mimics. (**D**,**E**) The mRNA and protein expression levels of Apaf-1, PARP, caspase-9, caspase-3, and cleaved cas3 were downregulated after treatment with propranolol, indicating that the rescue experiment was successful. * *p* < 0.05; ** *p* < 0.01; *** *p* < 0.001; ns means no statistical significance.

**Figure 4 biomedicines-13-01092-f004:**
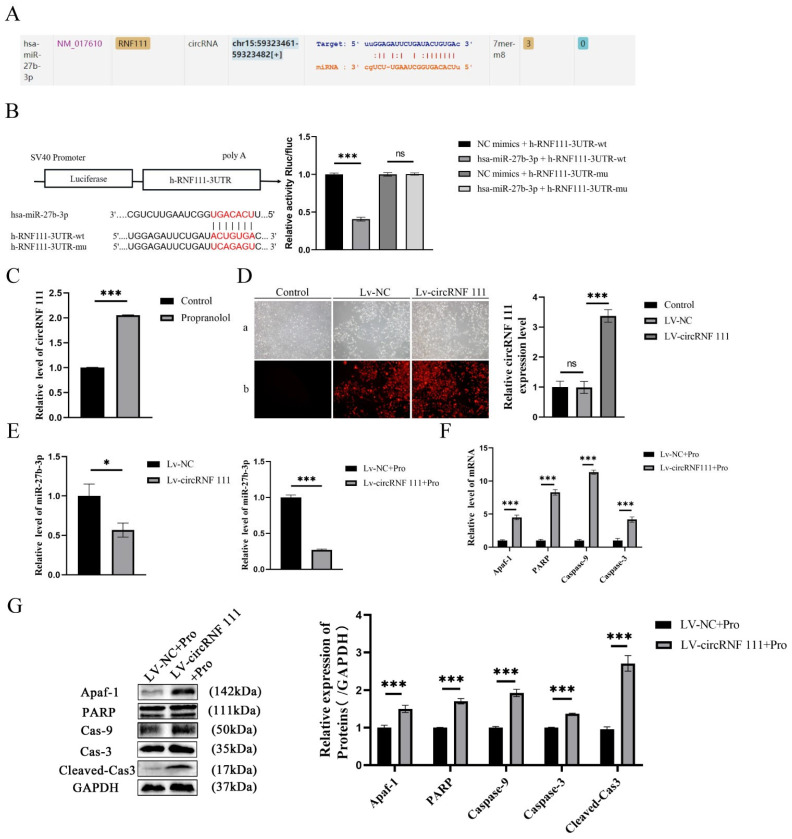
circRNF 111 reduces the expression of miR-27b-3p. (**A**) The ENCORI database predicts that circRNF111 has a complementary binding domain to miR-27b. (**B**) Luciferase reporter gene analysis showed that the co-transfection of the miR-27b-3p plasmid with the circRNF 111-wt plasmid significantly impaired luciferase activity. (**C**) The qRT-P CR outcome indicates that circRNF 111 expression was upregulated after being treated with propranolol. (**D**) Targeted relationship between miR-27b-3p and circRNF 111. Compared with the NC group, he-miR-27b-3p significantly downregulated the luciferase expression of h-RNF111-3UTR-WT (a: light mirror; b: fluorescence microscope), 4× total magnification. (**E**) qRT-PCR shows that compared with the LV-NC group, the expression of miR-27b-3p in the LV-circRNF 111 group was further downregulated after propranolol treatment. (**F**,**G**) After overexpression of miR-27b-3p, the mRNA and protein expression levels of Apaf-1, PARP, caspase-9, caspase-3, and cleaved cas3 were upregulated. * *p* < 0.05; ** *p* < 0.01; *** *p* < 0.001; ns means no statistical significance.

**Table 1 biomedicines-13-01092-t001:** Primer sequences for RT-PCT.

The Name of the Gene	Primer Sequence
*miRNA-27b-3P* *circRNF 111*	5′-AGTCCCGTGTCTGTAATGCC-3′ F: 5′-TAGCAGTTCCCCAATCCTTG-3′ R: 5′-CACAAATTCCCATCATTCCC-3′
*Apaf-1*	F: 5′-CGGCCCTGCGCATCTGATTCAT-3′ R: 5′-GGGCGAACGACTAAGCGGGACAG-3′
*PARP*	F: 5′-GGTCTTCCCCTACCCTCTCAA-3′ R: 5′-CGTTGTGTGTTCGCCTCT-3′
*Caspase-9*	F: 5′-CTGAGCCAGATGCTGTCCCATA-3′ R: 5′-GACACCATCCAAGGTCTCGATGTA-3′
*Caspase-3* *U6*	F: 5′-GACTGCGGTATTGAGACAGA-3′ R: 5′-CGAGTGAGGATGTGCATGAA-3′ F: 5′-CTCGCTTCGGCAGCACATATACTA-3′ R: 5′-ACGAATTTGCGTGTCATCCTTGCG-3′
*GAPDH*	F: 5′-TGTTCGTCATGGGTGTGAACC-3′ R: 5′-ATGGACTGTGGTCATGAGTCC-3′

## Data Availability

The data presented in this study are available upon request of the corresponding author due to restrictions imposed by the Institutional Review Board, which approved the study protocol.

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
