# Peer review of "MiR-27b-3p Reduces the Efficacy of Propranolol in the Treatment of Infantile Hemangioma by Inhibiting the Expression of Apaf-1"

_biomedicines, 2025, doi:10.3390/biomedicines13051092_

Round 1
Reviewer 1 Report
Comments and Suggestions for Authors
Relevance and novelty:
Infantile hemangioma (IH) is the most common benign vascular tumor in infants and children, with an incidence ranging from approximately 3% to 10%. It typically appears at birth or within the first month of life.
In their study, the Authors aimed to explore the role and mechanism of miR-27b-3p in treating infantile 13 hemangioma (IH) with Propranolol, with special focus on calrification of the cause of poor efficacy of Propranolol 14 in IH threatment, which brings some novelty into the science. The subject is worth of exploration and brings new lights on the IH potential treatment and falls into the Journal’s scope.
The methodology is well assigned, the structure of the manuscript is clear, Abstract summarises the the paper in the right structured way, however I would have some remarks.
Minor:
In Introduction, line 58-60 authors stated that potential mechanism of miR-27b-3p activity is not clear. Authors may wish to dicuss these findings in the paper below, and potentially revise their cathegoric statement:
https://doi.org/10.1016/j.abb.2020.108404
miR-27b-3p is implicated in reducing the malignancy of hemangioma-derived endothelial cells. IL-10 upregulates miR-27b-3p, which inhibits PCNA (a marker of cell proliferation) by destabilizing its mRNA and suppressing the transcription factor YY1. This pathway limits endothelial cell proliferation in IH.
Language: Language is mostly fine, needs “fine-tuning” however there are some minor faults or typos, like:
Fig 1A – Axis x – it should be Concentration; Fig 1E – Axis y – it should be: Relative level of mRNA, Fig 1E – Axis y – also should it be: Relative…
Fig 2A – Axis y - it should be: Relative level…; Fig 2B – Axis y – Relative level…; Fig 2F – Axis y – Relative level…; Fig 2G – Axis y – Relative…;
Fig 3C – Axis y – Relative… Fig 3D – Axis y – Relative…; Fig 3E – Axis y - Relative
Exactly the same with Fig. 4 (Relative level…)
Results and data presentation could be improved by enlarging some figures, like e.g.: Fig 3A, 3B, or even splitting/separating them from the rest.
Discussion
Discussion needs some additional work.
Lines 320-342 This part of the discussion is well written, but the literature references are missing altogether. A reference is needed for each statement.
Only 331-340: please rephrase to maintain continuity of reasoning
Do not repeat the results, rather discuss with other studies, example: https://www.nature.com/articles/s41598-020-60025-2
Please add limitations of this study.
Comments on the Quality of English Language
English could be improved.
Author Response
Comments to the Author:
- In Introduction, line 58-60 authors stated that potential mechanism of miR-27b-3p activity is not clear. Authors may wish to dicuss these findings in the paper below, and potentially revise their cathegoric statement:
https://doi.org/10.1016/j.abb.2020.108404
miR-27b-3p is implicated in reducing the malignancy of hemangioma-derived endothelial cells. IL-10 upregulates miR-27b-3p, which inhibits PCNA (a marker of cell proliferation) by destabilizing its mRNA and suppressing the transcription factor YY1. This pathway limits endothelial cell proliferation in IH.
Author Comments and Response: Thank you for your suggestion. The article is mostly about the potential mechanism of miR-27b-3p involved in developing infantile hemangiomas. Following sentences have been added: “It has been reported that miR-27b-3p is associated with reducing the malignancy of hemangioma-derived endothelial cells. IL-10 upregulates miR-27b-3p, which inhibits PCNA (a marker of cell proliferation) by destabilizing its mRNA and inhibiting the transcription factor YY1. This pathway limits endothelial cell proliferation in IH.”
Reference:
- Zeng, Z.; Chen, H.; Cai, J.; Huang, Y.; Yue, J., IL-10 regulates the malignancy of hemangioma-derived endothelial cells via regulation of PCNA. Archives of Biochemistry and Biophysics 2020, 688.
- Language: Language is mostly fine, needs “fine-tuning” however there are some minor faults or typos, like:
Fig 1A – Axis x – it should be Concentration; Fig 1E – Axis y – it should be: Relative level of mRNA, Fig 1E – Axis y – also should it be: Relative…
Fig 2A – Axis y - it should be: Relative level…; Fig 2B – Axis y – Relative level…; Fig 2F – Axis y – Relative level…; Fig 2G – Axis y – Relative…;
Fig 3C – Axis y – Relative… Fig 3D – Axis y – Relative…; Fig 3E – Axis y - Relative
Exactly the same with Fig. 4 (Relative level…)
Results and data presentation could be improved by enlarging some figures, like e.g.: Fig 3A, 3B, or even splitting/separating them from the rest.
Author Comments and Response: Thank you very much for your care. We have fixed the typos in all the words and figures.
- Discussion needs some additional work.
Lines 320-342 This part of the discussion is well written, but the literature references are missing altogether. A reference is needed for each statement.
Author Comments and Response: Thanks for your good suggestions! We have added some references to this part in lines 320-342, and the revised part is as follows: “For example, circRNA _ 0025202 regulates tamoxifen sensitivity and tumorigenesis in breast cancer by modulating the miR-182-5p / FOXO3a axis; as a miR-326 sponge, circ _ 0000515 promotes cervical cancer progression by up-regulating ELK11,2. Circ-RNF111 aggravates gastric cancer malignancy through mir-876-3p-dependent KLF12 regulation3. CircAP2A2 adsorbs miR-382-5p and regulates VEGFA expression to participate in the progression of infantile hemangioma4. However, there are few studies on ceRNA in IH, and the mechanism of ceRNA regulating IH is still in the exploratory stage. As a noncoding RNA, miRNA plays an important role in laryngeal cancer and inhibits norepinephrine-induced cardiomyocyte apoptosis by regulating the expression of Apaf-15. In recent years, studies have reported that miRNA can regulate angiogenesis, including vascular sprouting, vascular endothelial cell proliferation, survival, migration, and recruitment of vascular progenitor cells6. By controlling the expression of angiopoietin and promoting hemangioma endothelial cell apoptosis involved in the maturation of new blood vessels. miR-27b-3p is a subtype of miR-27 family, which is involved in the development of gastric cancer, glioma, breast cancer, and other diseases7-9. MiR-27b-3p can promote the apoptosis of hypoxic neurons by inhibiting Apaf-1, and this inhibition gradually decreases with age, similar to some infantile hemangiomas that gradually fade with age. CircRNA is also a functional regulator in regulating cell differentiation, angiogenesis, immune response, inflammatory response, and carcinogenesis10. Interestingly, a recent study found that circRNA is up-regulated under hypoxic or normoxic conditions and controls endothelial cell nodule formation and spherical budding11. CircRNF111 is considered to be closely related to the occurrence and development of breast cancer, colorectal cancer, and gastric cancer12.”
Reference:
- Sang, Y.; Chen, B.; Song, X.; Li, Y.; Liang, Y.; Han, D.; Zhang, N.; Zhang, H.; Liu, Y.; Chen, T.; Li, C.; Wang, L.; Zhao, W.; Yang, Q., circRNA_0025202 Regulates Tamoxifen Sensitivity and Tumor Progression via Regulating the miR-182-5p/FOXO3a Axis in Breast Cancer. MOLECULAR THERAPY 2019, 27 (9), 1638-1652.
- Tang, Q.; Chen, Z.; Zhao, L.; Xu, H., Circular RNA hsa_circ_0000515 acts as a miR-326 sponge to promote cervical cancer progression through up-regulation of ELK1. AGING-US 2019, 11 (22), 9982-9999.
- Wu, G.; Zhang, A.; Yang, Y.; Wu, D., Circ-RNF111 aggravates the malignancy of gastric cancer through miR-876-3p-dependent regulation of KLF12. WORLD JOURNAL OF SURGICAL ONCOLOGY 2021, 19 (1).
- Yuan, X.; Xu, Y.; Wei, Z.; Ding, Q., CircAP2A2 acts as a ceRNA to participate in infantile hemangiomas progression by sponging miR-382-5p via regulating the expression of VEGFA. JOURNAL OF CLINICAL LABORATORY ANALYSIS 2020, 34 (7).
- Sun, X.; Liu, B.; Zhao, X.-D.; Wang, L.-Y.; Ji, W.-Y., MicroRNA-221 accelerates the proliferation of laryngeal cancer cell line Hep-2 by suppressing Apaf-1. ONCOLOGY REPORTS 2015, 33 (3), 1221-1226.
- Shi, L.; Fisslthaler, B.; Zippel, N.; Froemel, T.; Hu, J.; Elgheznawy, A.; Heide, H.; Popp, R.; Fleming, I., MicroRNA-223 Antagonizes Angiogenesis by Targeting β1 Integrin and Preventing Growth Factor Signaling in Endothelial Cells. CIRCULATION RESEARCH 2013, 113 (12), 1320-+.
- Shen, S.-J.; Song, Y.; Ren, X.-Y.; Xu, Y.-L.; Zhou, Y.-D.; Liang, Z.-Y.; Sun, Q., MicroRNA-27b-3p Promotes Tumor Progression and Metastasis by Inhibiting Peroxisome Proliferator-Activated Receptor Gamma in Triple-Negative Breast Cancer. FRONTIERS IN ONCOLOGY 2020, 10.
- Zhang, Q.; Shao, W.; Li, Y.; Liu, L.; Chen, W.; Wang, C.; Li, B., Long non-coding RNA LINC01128 affects proliferation, migration, and invasion of glioma cells by regulating miR-27b-3p. FOLIA NEUROPATHOLOGICA 2022, 60 (3), 338-345.
- Cui, Y.; Huang, S.; Cao, J.; Ye, J.; Huang, H.; Liao, D.; Yang, Y.; Chen, W.; Pu, R., Combined targeting of vascular endothelial growth factor C (VEGFC) and P65 using miR-27b-3p agomir and lipoteichoic acid in the treatment of gastric cancer. JOURNAL OF GASTROINTESTINAL ONCOLOGY 2021, 12 (1), 121-132.
- Cheng, L.; Liu, Z.; Xia, J., New insights into circRNA and its mechanisms in angiogenesis regulation in ischemic stroke: a biomarker and therapeutic target. MOLECULAR BIOLOGY REPORTS 2023, 50 (1), 829-840.
- Zeng, Z.; Zhao, Y.; Chen, Q.; Zhu, S.; Niu, Y.; Ye, Z.; Hu, P.; Chen, D.; Xu, P.; Chen, J.; Hu, C.; Hu, Y.; Xu, F.; Tang, J.; Wang, F.; Han, S.; Huang, M.; Wang, C.; Zhao, G., Hypoxic exosomal HIF-1α-stabilizing circZNF91 promotes chemoresistance of normoxic pancreatic cancer cells via enhancing glycolysis. ONCOGENE 2021, 40 (36), 5505-5517.
- Wang, Z.; Jiang, Z.; Zhou, J.; Liu, Z., circRNA RNF111 regulates the growth, migration and invasion of gastric cancer cells by binding to miR-27b-3p. INTERNATIONAL JOURNAL OF MOLECULAR MEDICINE 2020, 46 (5), 1873-1885.
- Only 331-340: please rephrase to maintain continuity of reasoning
Do not repeat the results, rather discuss with other studies, example: https://www.nature.com/articles/s41598-020-60025-2
Author Comments and Response: Thank you for your suggestion. The article is mostly about the mechanism of propranolol involved in developing infantile hemangiomas. Following sentences have been added: “What’s more, it has been confirmed that hypoxia is involved in the development of IH, and that hypoxia is driven by HIF-2α. It is well known that HIF-2α levels are regulated by inflammation, differentiation, and stress signaling1. If we can develop specific targeting agents for any step of the pathway, then propranolol will have a better effect on the treatment of hemangioma.
Reference: Gomez-Acevedo, H.; Dai, Y.; Strub, G.; Shawber, C.; Wu, J. K.; Richter, G. T., Identification of putative biomarkers for Infantile Hemangiomas and Propranolol treatment via data integration. SCIENTIFIC REPORTS 2020, 10 (1).
- Please add limitations of this study.
Author Comments and Response: Thank you for your constructive advice. Following sentences have been added: “However, the limitation of our research is the lack of in vivo experiments. The conclusion would be more accurate if there were in vivo experiments. The last but not the least, propranolol treatment is now the first drug of choice for IH. However, lesion relapse after therapy remains an open question. To convincingly answer the question more research is warranted at the population level that can be further integrated and guide personalized treatment for children who experience side effects or are more susceptible to propranolol.”
Reviewer 2 Report
Comments and Suggestions for Authors
MiR-27b-3p reduces the efficacy of propranolol in the treatment of infantile hemangioma by inhibiting the expression of Apaf-1
The study Sijia Na et al., provide valuable insights into the molecular mechanisms underlying propranolol resistance in infantile hemangioma, focusing on the miR-27b-3p/Apaf-1 axis. The manuscript employs a comprehensive set of techniques, including CCK-8, TUNEL, flow cytometry, RT-qPCR, Western blot, and luciferase assays, to validate the hypothesis. 3. The introduction builds a clear rationale for the study, and the results follow a structured, logical sequence.
However, despite these merits I think this manuscript suffers from a several flaws. For starters, intro sections are dense and contain long, complex sentences. I’d consider breaking them down for better readability.
Cell viability assays should include normalization controls to ensure that the effects observed are specific. Please consider knockdown validation experiments for miR-27b-3p and Apaf-1 using siRNA or CRISPR to confirm specificity.
The discussion does effectively link findings to existing literature, but more emphasis could be placed on clinical implications.
Figures should be cross-referenced more explicitly in the text (for example Fig. 1). The axis labels in figures should be clear, with proper units. Statistical significance should be consistently reported. The concentration of propranolol -30 µM is well justified, but a dose-response curve showing IC50 determination would strengthen the argument.
Fig. 3, is not clear, is it possible to provide the raw blot images?
Same, please provide raw images and make sure the images are clearly visualized.
I think the potential limitations such as in vitro nature of the study, lack of in vivo validation etc should be acknowledged. I would consider discussing possible therapeutic strategies targeting miR-27b-3p to enhance propranolol efficacy.
Finally, minor grammatical errors need correction.
Please ensure consistent formatting of gene/protein names.
The reference formatting should be standardized according to the journal guidelines.
What is VEGFA?
I believe this study presents novel and relevant findings, and with improvements in clarity, data representation and in-depth discussion, it has a strong potential for publication.
Best of luck and keep up the good work,
Cheers!
Comments on the Quality of English Language
MiR-27b-3p reduces the efficacy of propranolol in the treatment of infantile hemangioma by inhibiting the expression of Apaf-1
The study Sijia Na et al., provide valuable insights into the molecular mechanisms underlying propranolol resistance in infantile hemangioma, focusing on the miR-27b-3p/Apaf-1 axis. The manuscript employs a comprehensive set of techniques, including CCK-8, TUNEL, flow cytometry, RT-qPCR, Western blot, and luciferase assays, to validate the hypothesis. 3. The introduction builds a clear rationale for the study, and the results follow a structured, logical sequence.
However, despite these merits I think this manuscript suffers from a several flaws. For starters, intro sections are dense and contain long, complex sentences. I’d consider breaking them down for better readability.
Cell viability assays should include normalization controls to ensure that the effects observed are specific. Please consider knockdown validation experiments for miR-27b-3p and Apaf-1 using siRNA or CRISPR to confirm specificity.
The discussion does effectively link findings to existing literature, but more emphasis could be placed on clinical implications.
Figures should be cross-referenced more explicitly in the text (for example Fig. 1). The axis labels in figures should be clear, with proper units. Statistical significance should be consistently reported. The concentration of propranolol -30 µM is well justified, but a dose-response curve showing IC50 determination would strengthen the argument.
Fig. 3, is not clear, is it possible to provide the raw blot images?
Same, please provide raw images and make sure the images are clearly visualized.
I think the potential limitations such as in vitro nature of the study, lack of in vivo validation etc should be acknowledged. I would consider discussing possible therapeutic strategies targeting miR-27b-3p to enhance propranolol efficacy.
Finally, minor grammatical errors need correction.
Please ensure consistent formatting of gene/protein names.
The reference formatting should be standardized according to the journal guidelines.
What is VEGFA?
I believe this study presents novel and relevant findings, and with improvements in clarity, data representation and in-depth discussion, it has a strong potential for publication.
Best of luck and keep up the good work,
Cheers!
Author Response
Comments to the Author:
- Cell viability assays should include normalization controls to ensure that the effects observed are specific. Please consider knockdown validation experiments for miR-27b-3p and Apaf-1 using siRNA or CRISPR to confirm specificity.
The discussion does effectively link findings to existing literature, but more emphasis could be placed on clinical implications.
Author Comments and Response: Thank you for your constructive suggestion. we will do our best to address your suggestions below. At the end of the article, we emphasize the clinical significance of this study for infantile hemangioma. Following sentences have been added: “This study reveals that Propranolol treats hemangiomas by activating the APAF1-mediated apoptotic pathway, while circRNF 111 induces propranolol-induced HUVECs apoptosis via miR-27b-3p/Apaf-1. If there is a targeted inhibitor of miR-27b-3p, propranolol can be promoted to induce apoptosis by reducing the expression of miR-27b-3p, which has a great role in the treatment of hemangioma. This is also the first report on the role of miR-27b-3p in the treatment of hemangioma with propranolol. These results will deepen the understanding of propranolol in the treatment of infantile hemangioma, and provide a theoretical basis for the sensitization effect of Propranolol in treating hemangioma.”
- Figures should be cross-referenced more explicitly in the text (for example Fig. 1). The axis labels in figures should be clear, with proper units. Statistical significance should be consistently reported. The concentration of propranolol -30 µM is well justified, but a dose-response curve showing IC50 determination would strengthen the argument.
Author Comments and Response: Thank you for your advice! We have replaced the figures with higher definition and resolution.
- Fig. 3, is not clear, is it possible to provide the raw blot images? Same, please provide raw images and make sure the images are clearly visualized. I think the potential limitations such as in vitro nature of the study, lack of in vivo validation etc should be acknowledged. I would consider discussing possible therapeutic strategies targeting miR-27b-3p to enhance propranolol efficacy.
Author Comments and Response: Thank you for your good suggestion. We have uploaded all raw western blot images. And we added the potential limitations and therapeutic strategies targeting miR-27b-3p to enhance propranolol efficacy in the discussion part. Following sentences have been added: “However, the limitation of our research is the lack of in vivo experiments. The conclusion would be more accurate if there were in vivo experiments. The last but not the least, propranolol treatment is now the first drug of choice for IH. However, lesion relapse after therapy remains an open question. To convincingly answer the question more research is warranted at the population level that can be further integrated and guide personalized treatment for children who experience side effects or are more susceptible to propranolol.”. “If there is a targeted inhibitor of miR-27b-3p, propranolol can be promoted to induce apoptosis by reducing the expression of miR-27b-3p, which has a great role in the treatment of hemangioma. This is also the first report on the role of miR-27b-3p in the treatment of hemangioma with propranolol. These results will deepen the understanding of propranolol in the treatment of infantile hemangioma, and provide a theoretical basis for the sensitization effect of Propranolol in treating hemangioma.”
- Finally, minor grammatical errors need correction. Please ensure consistent formatting of gene/protein names. The reference formatting should be standardized according to the journal guidelines. What is VEGFA?
Author Comments and Response: Thank you for your suggestion. We have revised the format of all gene/protein names and revised the format of all references. VEGFA refers to vascular endothelial growth factor A. It plays a key role in pathological angiogenesis and angiogenesis of IH.

Reviewer 3 Report
Comments and Suggestions for Authors
It is an interesting manuscript but i have few comments:
- In Materials and Methods : Line 78: Why did you use human umbilical vein endothelial cells ? and also where is the reference of Cell Culture and Treatment?
- In Statistical Analysis: Line 173: SPSS Version 22, need to be update
Author Response
Comments to the Author:
- In Materials and Methods: Line 78: Why did you use human umbilical vein endothelial cells? And also where is the reference of Cell Culture and Treatment?
Author Comments and Response: Our research is about hemangiomas, so we need to look for cells associated with blood vessel. It has been reported that VEGF and its receptor (vascular endothelial growth factor receptor; VEGFR) is critical in the angiogenesis of infantile hemangiomas. VEGFR2, one of the three receptors of VEGF, is a critical receptor for blood vasculature development, and infantile hemangiomas are induced by elevated VEGF signaling through VEGFR2. Moreover, VEGFR2 has been previously targeted to promote urea-encapsulated liposomes to target hemangioma vascular endothelial cells. Thus, we use human umbilical vein endothelial cells to finish our experiment. HUVECs were purchased from Sebacon. There will be detailed cell culture and treatment methods in the protocol. Moreover, some references refer to the cell culture methods as follows:
References:
- Li, H.; Teng, Y.; Xu, X.; Liu, J., Enhanced rapamycin delivery to hemangiomas by lipid polymer nanoparticles coupled with anti-VEGFR antibody. INTERNATIONAL JOURNAL OF MOLECULAR MEDICINE 2018, 41 (6), 3586-3596.
- Sohn, C.-H.; Park, S. P.; Choi, S. H.; Park, S.-H.; Kim, S.; Xu, L.; Kim, S.-H.; Hur, J. A.; Choi, J.; Choi, T. H., MRI molecular imaging using GLUT1 antibody-Fe3O4 nanoparticles in the hemangioma animal model for differentiating infantile hemangioma from vascular malformation. NANOMEDICINE-NANOTECHNOLOGY BIOLOGY AND MEDICINE 2015, 11 (1), 127-135.
- In Statistical Analysis: Line 173: SPSS Version 22, need to be update.
Author Comments and Response: Thank you for your suggestion. We have updated the SPSS Version 22 to SPSS Version 27.
Round 2
Reviewer 2 Report
Comments and Suggestions for Authors
Thank you very mu’ch for addressing all the comments and concerns in a positive spirit. In the first round of reviews, authors were requested to provide raw images for the blots but it doesn’t seem the case. Blots are not clear especially Fig 2 G. Please provide clear for bots for the aesthetics of the panels.
Rest seems fine. Good luck!
Author Response
Comments to the Author:
- In the first round of reviews, authors were requested to provide raw images for the blots but it doesn’t seem the case. Blots are not clear especially Fig 2 G. Please provide clear for bots for the aesthetics of the panels.
Author Comments and Response: Thank you for your suggestion. We have uploaded all raw western blot images and adjusted Figure 2 G to be clearer. The modified data graph is as follows.
